# GenSim2:
# Scaling Robot Data Generation with Multi-modal and Reasoning LLMs

**Pu Hua**[1*] **Minghuan Liu**[2,3*] **Annabella Macaluso**[2*] **Yunfeng Lin**[3]
**Weinan Zhang**[3] **Huazhe Xu**[1] **Lirui Wang**[4‡]
Institute for Interdisciplinary Information Sciences, Tsinghua University[1],
UCSD[2], Shanghai Jiao Tong University[3], MIT CSAIL[4]
https://gensim2.github.io/

**Abstract:** Robotic simulation today remains challenging to scale up due to the human efforts required to create diverse simulation tasks and scenes. Simulation-trained policies also face scalability issues as many sim-to-real methods focus on a single task. To address these challenges, this work proposes `GenSim2`, a scalable framework that leverages coding LLMs with multi-modal and reasoning capabilities for complex and realistic simulation task creation, including long-horizon tasks with articulated objects. To automatically generate demonstration data for these tasks at scale, we propose planning and RL solvers that generalize within object categories. The pipeline can generate data for up to 100 articulated tasks with 200 objects and reduce the required human efforts. To utilize such data, we propose an effective multi-task language-conditioned policy architecture, dubbed proprioceptive point-cloud transformer (`PPT`), that learns from the generated demonstrations and exhibits strong sim-to-real zero-shot transfer. Combining the proposed pipeline and the policy architecture, we show a promising usage of `GenSim2` that the generated data can be used for zero-shot transfer or co-train with real-world collected data, which enhances the policy performance by 20% compared with training exclusively on limited real data.

**Keywords:** Coding Large Language Models, Vision Language Models, Robotic Simulation, Multi-task Sim-to-Real Transfer, Articulated Object Manipulation

## 1 Introduction

Robot learning requires large amounts of interaction data and evaluation, which are expensive to acquire at scale in the real world. Robot simulation holds the promise of providing such data and verification in high diversity and efficiency across objects, tasks, and scenes. While the ability to simulate has led to many successes in AI across Gaming, Go, and Mathematical Proofs [2, 3, 4], there are two requirements for such a path to be successful in robotics: The data needs to scale in *complexity* without significant human efforts and the data needs to be *realistic* enough to transfer to the real world. Previous works [5, 6, 7, 8, 9, 10, 11] have made significant progress in scalable simulation benchmarks in robotics and training policies on the simulation data.

Foundation models [12], particularly generative models pre-trained on internet-scale data [13, 14, 15], have demonstrated impressive capabilities required for generating robot simulation tasks, such as coding [16], spatial reasoning [17], task semantics [9], planning [18, 19], video prediction[20, 21], and cost and reward understanding [22, 23]. While foundation models have shown impressive capabilities to output actions to solve robotic tasks directly in the real world [24], simulation provides a low-cost and scalable platform to learn robust end-to-end policies. In addition to generating numer-

---

∗ equal contribution. ‡ project lead.
8th Conference on Robot Learning (CoRL 2024), Munich, Germany

**(A) Large-scale Task and Data Generation**

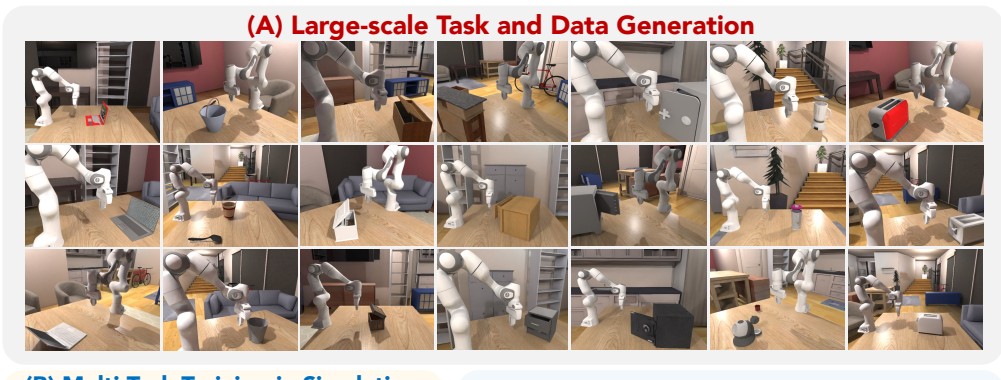

**(B) Multi-Task Training in Simulation**

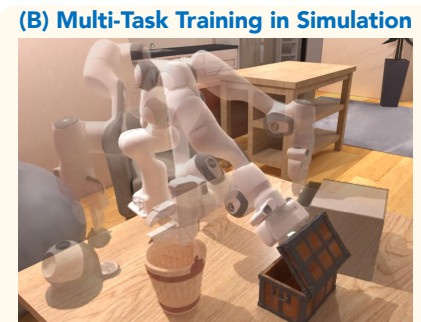

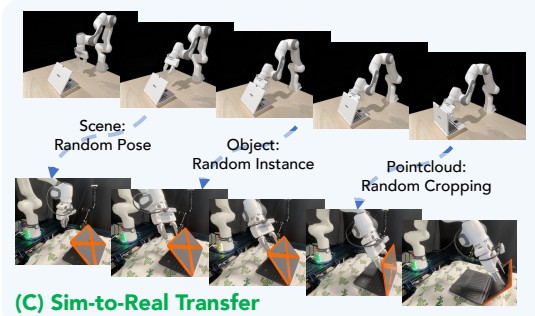

Scene: Random Pose     Object: Random Instance     Pointcloud: Random Cropping

**(C) Sim-to-Real Transfer**

*Figure 1.* `GenSim2` introduces a scalable task and data generation framework in SAPIEN [1] for articulation objects with multi-modal and reasoning LLMs. The framework comprises three main stages: 1) *(Top)* We first generate large-scale robotics tasks and collect massive data with LLMs; 2) *(Bottom-left)* Then, we train a multi-task point cloud-based policy in simulation with imitation learning; 3) *(Bottom-right)* Finally, we zero-shot transfer the policy to the real world.

ous tasks, automatically solving them to produce demonstration data and transferring the learned skills to the real world present significant challenges. To address this, the data generation and policy learning pipelines must scale efficiently while minimizing the sim-to-real gaps.

In this work, we propose `GenSim2` (Fig. 1), a scalable framework that amplifies robotics data by generating diverse articulated and long-horizon simulation tasks and demonstrations. In addition to the semantic knowledge and compositional capabilities in language models, these tasks also require extended 6-dof motions with contacts. To generate demonstrations for such tasks, we develop various solvers, including a keypoint-based motion planner that generalizes across object categories and initial conditions. Moreover, we use the visual information in a multi-modal LLM (MLLM) to iteratively generate and verify novel tasks, to the extent of 100 tasks to train a multi-task policy.

`GenSim2` tackles more challenging articulated and long-horizon tasks beyond top-down pick-and-place [9]. These complex tasks require grounding and reasoning capabilities beyond previous LLM techniques. Therefore, `GenSim2` is an agent pipeline featuring multi-modal foundation models (e.g. GPT-4V) and reasoning models (e.g. OpenAI o1) as well as generalized keypoint planners for such complex tasks. To fully utilize the generated data, we further propose a sim-to-real native policy architecture, dubbed proprioceptive point-cloud transformer (PPT), with point-cloud observations and language conditioning. This effective architecture is designed for sim-to-real transfer with the large-scale data generated in simulation.

In our experiments, we have generated over 100 articulated object manipulation tasks with 35 objects (over 200 instances), including 50 long-horizon tasks. Our task generation pipeline achieves 25% better success rates than previous works. After using an automated pipeline to solve these tasks and generate task demonstrations, our multi-task policy can jointly solve 24 tasks with random scene configurations and only suffer a 3% performance drop when tested on unseen object instances. More importantly, by integrating the `GenSim2` pipeline with the designed policy architecture, we achieve promising results across 8 real-world articulated object tasks, through such as zero-shot transfer and co-training with real-world data. This approach leads to a 20% improvement in policy performance compared to training exclusively on limited teleoperation data.

In summary, we show a promising way to reduce data collection efforts and solve real-world problems through massive simulated data generation with MLLMs. Our contributions are as follows: (1)

We design a robotic simulation task generation pipeline that explores the usage of internet-scale visual and language knowledge embedded in coding multi-modal LLMs (e.g. GPT-4V), to generate up to 100 articulated object manipulation tasks. (2) We propose a simulated demonstration generation pipeline that can generate high-quality data with minimal labeling efforts. The solution integrates smooth motions from model-based planners that have spatial and object-level generalization. (3) We develop a novel policy architecture for seamless sim-to-real transfer that distills multi-task demonstration efficiently and with strong sim-to-real performance.

## 2    Related Work

**Task Generation in Robotic Simulation.** Training and evaluation for robotics have greatly benefited from developments in robotic simulation such as Mujoco, NVIDIA Isaac, Drake, and Sapien [25, 26, 27, 1]. With these tools, researchers and engineers have been able to develop benchmarks with tens to hundreds of unique, hand-designed robotic tasks [28, 29, 30, 31, 32, 33, 28]. However, the requirements of high-quality assets, scenes, and task design along with required verifications for task solvability and real-world transfer demand a significant amount of human skill and effort. Recent works have explored methods such as domain randomizations [34, 7, 35, 36], procedural asset generation [8, 37] and text-to-3D synthesis [38, 39, 40, 41] to mitigate some of these efforts. Researchers also investigated how generative models can generate suites of robotic or agent tasks [23, 42, 43, 44, 45, 9, 10, 11] or generate interactive simulations directly from videos [20, 21]. In particular, GenSim [9] develops a novel pipeline that generates over 100 simulation tasks utilizing LLMs, verifies these tasks, uses them to generate data, and trains multi-task policies on top of this data. Similarly, RoboGen [10] uses RL to solve complicated locomotion tasks and deformable manipulation and RoboCasa [11] generates massive high-fidelity tasks for multiple embodiments in mobile manipulation tasks. In our work, we leverage multi-modal LLMs to generate 6-DOF robotic manipulation and long-horizon tasks at scale. Furthermore, we focus on complex yet realistic tasks that can solved with different solvers and transferred to the real world.

**Multi-Task Policy Learning with 3D Information.** With the surge of robotic data, recent works have explored multi-task policy learning, usually conditioned on language inputs. In particular, policy learning often benefits from access to explicit 3D information when doing generalized 6-DOF tasks [46, 47]. [48] uses RGB-D inputs and output pixel-level affordance map. [46] uses voxel as policy input information and output keyframe actions. [47, 49] uses point cloud as inputs to the policies. [50] uses keypoints at the corners to manipulate boxes with dexterous hands. In our work, we focus on point-cloud transformers as multi-task policies for articulated object manipulation.

**Sim-to-Real Transfer.** While simulation provides scalable training data and evaluation runs, sim-to-real transfer [51, 52] is a challenge and an active research area. RL-based controllers [53] leverage simulation environments for large-scale interactions to learn robust behaviors. Previous works have also explored techniques such as domain randomizations [54, 34], domain adaptations [55], and policy composition [56] to improve sim-to-real transfer. Our work can be viewed as a distillation process from the knowledge in foundation models such as MLLMs[14, 13], into policies, which can have a smaller sim-to-real "semantic" gap compared to rule-based simulation data.

## 3    Generating Tasks and Training Policies at Scale

The proposed `GenSim2` framework executes a series of processes including *task proposal* (3.1), *demonstration generation* (3.2), and *policy learning and transfer* (3.3). We illustrate the pipeline in Fig. 2. Our work enhances GenSim [9] at the level of task complexity beyond top-down pick-and-place, which necessitates grounded multi-modal task designs, generic 6-dof motion planners and RL solvers, and scalable policy architectures.

### 3.1    Task Proposal

**Primitive Task.** Our pipeline begins with proposing primitive tasks, namely tasks with a single simple motion (e.g. opening a box), by prompting an LLM to generate a novel task, defined by its

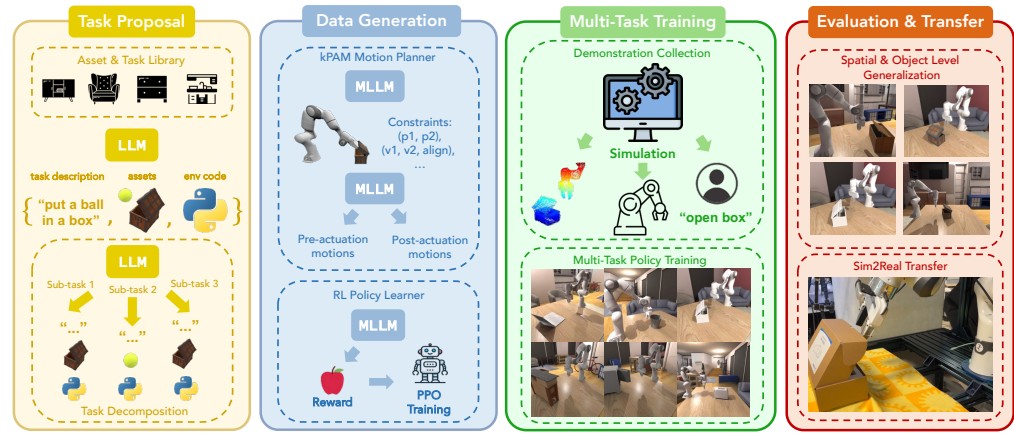

*Figure 2.* **Overview of `GenSim2` framework.** The pipeline consists of (1) task proposal, (2) solver creation, (3) multi-task training, and (4) generalization evaluation and sim-to-real transfer.

task description (a short phrase or sentence), used assets and task code. We start from a fixed asset library and a small task library initialized with hand-designed example tasks. We parse them into the prompts for in-context learning. We query the LLM to discover applicable assets for creating a new task. The LLM then takes in as input the generated task definition and outputs the corresponding task code. This code is finally compiled and ready for demonstration generation in Section 3.2.

**Long-horizon Task.** Moreover, we extend our pipeline to generate long-horizon tasks that consist of multiple steps of primitive tasks and involve manipulating articulated and rigid-body objects. Examples include opening a box, placing a ball inside, and then closing the box. When generating a long-horizon task, we execute a task decomposition process between task proposal and code implementation, to decompose the long-horizon task into several sub-tasks (primitive tasks). We use two distinct methods for generating long-horizon tasks: *(1) Top-down*: We directly generate a task in a curriculum, then decompose the task into several sub-tasks, each with a dedicated solver. *(2) Bottom-up*: We first generate primitive tasks and build up a task library. Then the LLM will be prompted to select tasks from the pre-built task library to compose a new task. We have observed reasoning LLM to improve task proposal performance in this stage. After we have finished the task planning, we continue to generate solvers and demonstrations for the subtasks in succession. Fig. 9 demonstrates visualizations of some generated long-horizon tasks.

## 3.2 Demonstration Generation

In this section, our goal is to create a solver to generate demonstrations given a task (or sub-task) code. The objective of our generation pipeline is to collect large-scale, high-quality data for general 6-DOF manipulation task learning in the real-world setting, our task solver should meet the following requirements: 1) Universal for 6-DOF tasks without task-specific designs; 2) Robust to different scene configurations; 3) Fast to execute; 4) Deterministic with high success rates. While such a solver is easy to design for top-down pick-and-place, in which we only need to specify some 2D position on the table as waypoints and execute primitive "pick" and "place" actions, it becomes challenging for more complex tasks such as general articulated object manipulation.

To address this challenge, we propose and investigate the planner config generation with a multi-modal LLM [14]. We also propose two types of task solvers: a kPAM-based motion planner and a complementary RL-based policy learner (see in Appendix A.3), to solve the generated tasks.

**Motion Planner.** kPAM[57] is a keypoint-based planner proposed for category-level manipulation tasks. It defines a robotic task by an *actuation pose*, namely the homogeneous transformation required to manipulate the target object, and it addresses an optimization problem based on several keypoint-based constraints to get this pose. Following Wang et al. [33], we improve and extend kPAM to produce an object-centric trajectory of end-effector poses, termed *actuation motions*, to solve a generated task. We parse the constraints and actuation motions into parameterized configurations that are easily interpretable and codable. We provide an illustration in Fig. 7 to show how the kPAM planner solves a task.

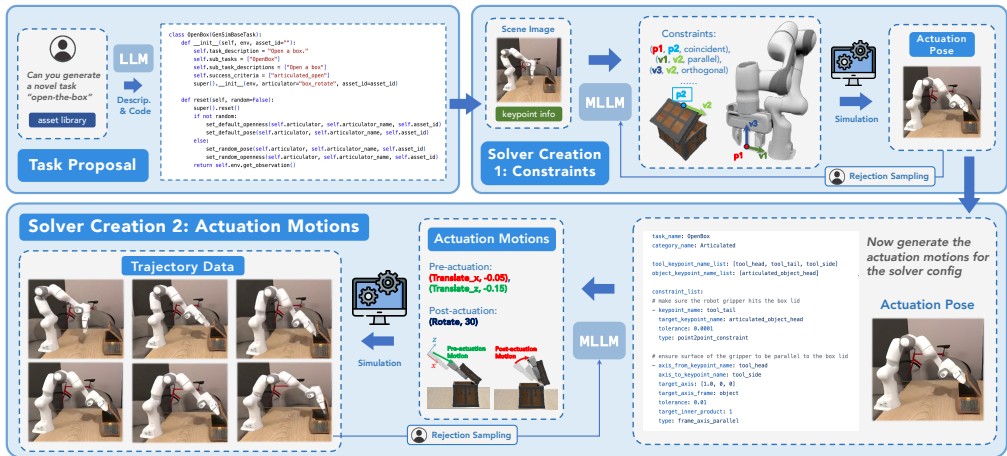

*Figure 3.* **Multi-modal Task Generation Pipeline.** `GenSim2` first is prompted to generate the task code, given few-shot examples and available assets. It then renders the scene image and the keypoint information of the task assets is fed into the MLLM model to generate a planner config for the actuation pose. The actuation pose is then extended to actuation motions, which are fed into GPT-4V for inspections. In this example, the pipeline produces a task motion for opening the box.

**Multi-modal Task Solver Generation.** Compared to previous works that primarily utilize language prompts, our approach incorporates visual information and key points [58, 59, 60], serving as an explicit representation of scene details (e.g., object structures, affordance and spatial information). The pipeline of solver generation is depicted in Fig. 7. First, we execute the generated code and capture scene images. This visual data, together with object key points, are then used to prompt an MLLM to generate the planner constraints. Subsequently, we visualize the actuation pose defined by these constraints. Finally, we query the MLLM to generate the actuation motions. Optionally, we also introduce reject sampling to guide the MLLM in refining its previous outputs.

### 3.3 Policy Learning and Tranfer

**Multi-Task Training.** We design a multi-task policy structure, denoted Proprioception Point cloud Transformer (PPT), as illustrated in Fig. 4. Specifically, we handle three kinds of observations: point cloud, proprioceptive states, and language task description, all of which can be obtained from the real world. Each observation is separately tokenized via respective encoders and cross-attention, fused together in the shared latent space through transformer blocks, and post-processed into global condition tokens [61]. Given the global condition tokens, the policy predicts a sequence of actions through the policy head. In our implementation, we test against various policy heads such as MLPs, the transformer decoder [62], and the diffusion model [63]. See Appendix B.2 for more details.

**Sim-to-Real Transfer.** To transfer between simulation and real environments, we ignore the color information of the point cloud, and augment the point cloud observations of simulated demonstrations during training. This augmentation involves cropping, adding Gaussian noise as perturbations, and randomly dropping points. In the real-world setting, we process the actual point cloud using uniform sampling, farthest point sampling, and outlier removal. This processed clean point cloud serves as input to the multi-task policy during inference.

## 4 Experiment: Task and Data Generation

In this section, we aim to verify the feasibility of the task generation framework and investigate the effectiveness of the task and data generation pipeline.

**Evaluation Criteria.** We evaluate the task generation procedure mainly on two types of success rates: *1) execution rate*, the success rate of completing the whole pipeline without any error such as syntax or runtime error; *2) solution rate*, the success rate of solving the generated task.

**Ablation Study.** In this section, we conduct a careful ablation study on each component of `GenSim2`. The results, depicted in Fig. 13, are summarized as follows:

*Types of LLMs*: We test multi-modal LLM, reasoning LLM, and vanilla LLM on solution rate in Fig. 5 left. Without visual data as input, the performance of the pipeline deteriorates with signifi-

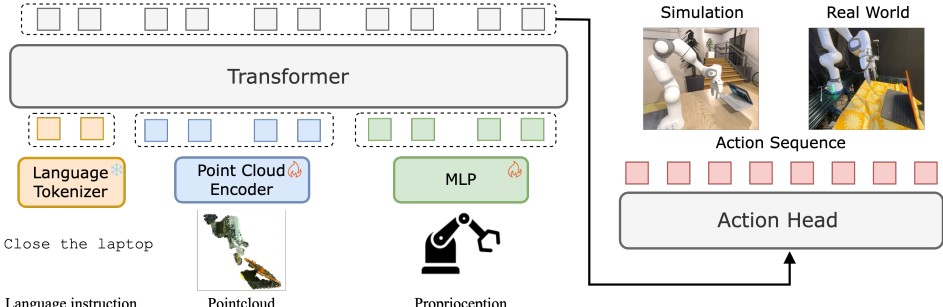

*Figure 4.* The proposed Proprioception Point-cloud Transformer (`PPT`) policy architecture maps language, point cloud, and proprioception inputs in a shared latent space for action prediction. The policy action head supports various architectures from diffusion [63] to transformer decoder [62].

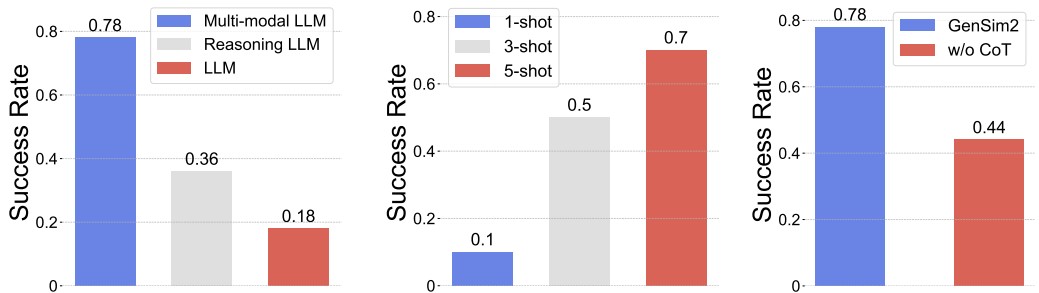

*Figure 5.* **Ablation study on components of our generation pipeline.** All results are based on no less than 10 generations. *Left)* We use various types of LLMs for solver generation and find that multi-modal LLM (GPT-4V) outperforms the others. *Middle)* We find that the performance of our method will increase, as we increase the maximum iteration for reject sampling. *Right)* We observe that splitting solver generation into a prompt chain surpasses generating the whole solver config.

cantly lower success rates in task generation under both criteria. The performance drops because the constraints for actuation pose can not be correctly generated as it requires details for object structure. Meanwhile, the actuation motions also suffer from large errors because the vanilla LLM has no access to spatial relationships between objects. This highlights the crucial role of visual information in the success of `GenSim2`, as it provides essential details about the scene such as spatial relationships and the appearance of objects. Moreover, reasoning capability also improves vanilla LLM's score on solver generation. We hypothesize that the reasoning trace of "think" before coding, based on the task and config structure can reduce hallucinations in generating kPAM constraints. See Appendix E.2 and E.3 for more details.

*Multi-shot Reject Sampling*: In Fig. 5 middle, we have visualized the solution rates of our pipeline under different maximum iterations of multi-shot rejection sampling. The results indicate that our pipeline with multi-shot rejection sampling surpasses the version without it, highlighting the importance of self-reflection in LLMs. Additionally, allowing LLMs more opportunities to refine their answers through iterations leads to improved performance. We also explore the use of rejection sampling from GPT-4V, which takes solver config visualizations as input and determines their effectiveness, in our pipeline. Due to the domain gaps of visual perception and understanding capabilities of MLLMs, particularly in processing 3D robotic scenes ([64, 65]), the MLLM can generate irrelevant responses. More powerful models and advanced prompting and finetuning methods are likely to improve these in future work.

*Chain-of-thought Prompt Design:* We observe that dividing solver generation into a prompt chain (first generating constraints and then actuation motions) yields better results (over 30%) than directly outputting the complete solver at once in Fig. 5 right. This approach allows for focused, sequential processing of components, leveraging outputs from earlier stages for subsequent ones.

**Comparison with Baseline.** We compare the pipeline of `GenSim2` using a kPAM motion planner with another open-sourced framework RoboGen[10], which encompasses a large number of long-horizon articulated tasks. For primitive tasks, we consider the sub-tasks within a long-horizon task

*Table 1.* Comparison with RoboGen on task generation success rates. **GenSim2-B** represents the bottom-up long-horizon task proposal method, while **GenSim2-T** and **GenSim2-T (o1)** represent top-down methods using GPT-4 and OpenAI-o1 respectively.

| Type | Primitive | | Long-horizon | | | |
|---|---|---|---|---|---|---|
| Method | GenSim2 | RoboGen | GenSim2-B | GenSim2-T | GenSim2-T (o1) | RoboGen |
| Execution | **0.94** | 0.94 | **1.00** | **0.83** | **0.87** | 0.76 |
| Solution | **0.78** | 0.58 | **0.68** | **0.54** | **0.60** | 0.43 |

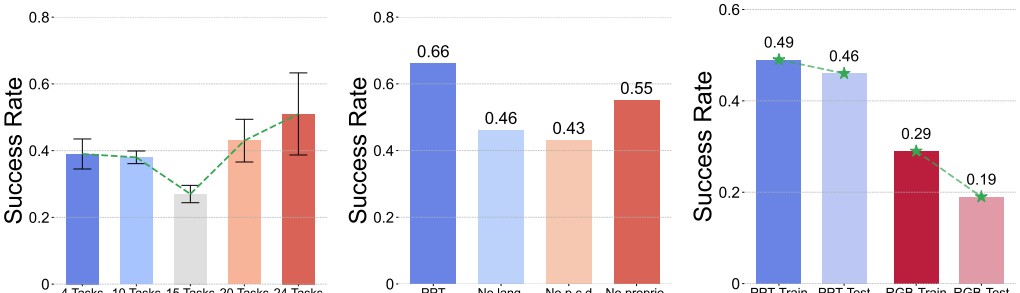

*Figure 6.* **Multi-task training results.** All results are evaluated with 20 episodes per task. *Left)* As the number of training tasks increases, the policy performance decreases first and then increases. *Middle)* We respectively ablate the input modalities leveraged in our policy architecture on 4 tasks. *Right)* We find that policies exhibit strong object-level generalization. For each architecture, the left and right bars illustrate the success rates during the training and testing phases, respectively.

in RoboGen for comparison. For long-horizon tasks, we compare our top-down methods (labeled as **GenSim2-T** and **GenSim2-T (o1)**, utilizing GPT-4 and OpenAI-o1 as task decomposers, respectively) and bottom-up method (**GenSim2-B**) with RoboGen's long-horizon tasks.

The results, presented in Tab. 1, indicate that our method exceeds the baseline in both primitive and long-horizon task settings. The underlying reason is our approach's implementation of fine-grained motion planner generation, whereas RoboGen directly generates a task-specific reward function and relies on reinforcement learning for task training, which can be inherently more fragile. Additionally, the bottom-up approach surpasses the top-down approach in both metrics. This advantage stems from the former's reliance on composing existing tasks, where most failures occur during the chaining of sub-tasks, whereas the latter necessitates generating each sub-task from scratch. Furthermore, the enhanced reasoning capabilities of OpenAI o1 contribute to more logical task decompositions compared to vanilla GPT-4, resulting in superior performance across both evaluation criteria.

## 5 Experiment: Multi-Task Policy Training and Transfer

In this section, we show how the generated data by GenSim2 can be used by multi-task imitation learning and sim-to-real transfer.

### 5.1 Multi-task Training in Simulation

**Training.** With the tasks and data generated by our pipeline, we train a multi-task policy (with 382M parameters) across different numbers of tasks and test its generalization to new scenarios. In Fig. 6 left, we jointly train a different number of LLM-generated tasks and test on 4 original tasks under the low data regime such as 10 demos per task. Interestingly, adding more tasks will first drop the performance and then increase it by virtue of scaling. Please refer to Appendix B for more details.

**Generalization.** After training, we test the generalization ability of our policy to unseen object instances, compared with a policy with RGB inputs. We split the instances into train/test sets. Evaluated results are shown in Fig. 6 right, where we find that the success rates of the PPT architectures only drop by less than 3% on unseen instances, whereas the RGB policies reduce by quite a bit. These results indicate that by generating data with object-level and spatial-level variance as domain randomization, along with the pointcloud-based policy architecture design, the trained policy can acquire generalization in both aspects, which lays the foundation for further sim-to-real transfer.

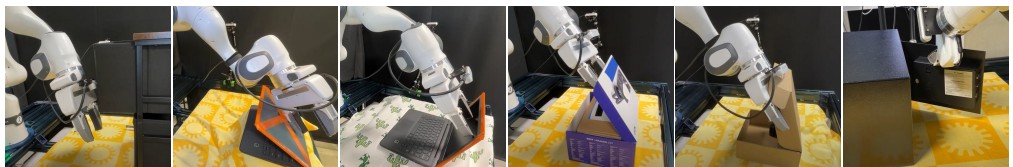

*Figure 7.* **Real Experiments:** We observe that the multi-task policy, trained on simulation data, can perform robustly on various unseen real-world objects through sim-to-real transfer.

*Table 2.* **Evaluation on a set of real-world tasks using one multi-task policy.** We report the per-task success rate of 10 episode evaluations with 8 real-world tasks. *Sim-only* represents policy trained purely with 100 simulation data generated by GenSim2. *Real-only* represents policy trained with 10 real-world data collected by teleoperation. *Combined* represents policy co-trained with combined simulation and real-world data.

| Training Data | OpenLaptop | CloseLaptop | OpenSafe | CloseSafe | CloseDrawer | SwingBucket | OpenBox | CloseBox | Average |
|---|---|---|---|---|---|---|---|---|---|
| *Real-only* | 0.5 | 0.0 | 0.2 | 0.4 | **1.0** | 0.5 | 0.2 | 0.1 | 0.363 |
| *Sim-only* | 0.7 | 0.5 | 0.1 | 0.3 | 0.8 | 0.5 | **0.5** | 0.0 | 0.425 |
| *Combined* | **0.8** | **0.7** | **0.3** | **0.6** | **1.0** | **0.8** | 0.0 | **0.4** | **0.575** |

## 5.2 Real-World Experiments

To evaluate the quality of the data collected in simulation and how GenSim2 helps in real-robot tasks, we test sim-to-real and co-training on the generated data over several tasks. As for the setup, we use the Franka Research 3 robot arm with a modified, deformable TPU parallel gripper for easier grasping. The robot work cell is equipped with three RealSense D435 cameras: one wrist-mounted and two externally facing the scene. Each captures an RGB-D observation which is combined and processed into a point cloud to be used as observations. More details are listed in Appendix C.

We perform experiments on 8 real-world tasks, collecting 100 demonstrations for each task in simulation, along with an additional 10 real-world demonstrations via teleoperation. We assess the quality and utility of the generated data across three distinct training setups: (1) using only simulation data, (2) using only real-world data, and (3) using a combination of both. The results of these evaluations are presented in Tab. 2. Our findings can be summarized as follows: (a) data generated by GenSim2 enables effective zero-shot sim-to-real transfer, with the resulting policy outperforming one trained solely on limited real-world data; and (b) when co-trained with real-world data, the data generated by GenSim2 significantly enhances the policy performance even by 20% in absolute values and 50% in relative scale. These results underscore the potential of large-scale, high-quality data generation, like GenSim2, to reduce the burden of extensive real-world data collection while improving policy effectiveness for real-world tasks.

## 6 Limitations and Discussions

Our proposed method has several limitations. Due to the lack of "robotic centric" knowledge, such as 3D spatial understanding, foundation models like GPT-4V still face hallucination issues in creating meaningful tasks and successfully coding them. Additionally, human involvement, though minimal, is still required to generate these complex manipulation tasks. Finally, we have only considered 6-dof tasks in zero-shot sim-to-real transfer with limited point cloud observations.

## 7 Conclusion and Future Work

In this work, we propose GenSim2, a task and demonstration generation framework that utilizes multi-modal foundation models to generate up to 100 robotic simulation tasks at scale, such as long-horizon manipulation with articulated objects. We ablate on different task solvers and generation components, as well as propose a multi-task pointcloud-based policy architecture that distills generated demonstrations from simulation transfer to the real world. Future works include expanding task complexity and diversity through advanced multi-modal agents and 3D asset generation. Future works can explore advanced sim-to-real methods for complex tasks with multiple embodiments.

## Acknowledgments

We want to thank Professor Xiaolong Wang for his kind support and discussion of this project. We thank Yuzhe Qin and Fanbo Xiang for their generous help in Sapien development. We thank Mazeyu Ji for his help with real-world experiments. This work is partly supported by the Amazon Greater Boston Tech Initiative and Amazon PO No. 2D-06310236.

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

# Appendix

## A  Task Generation Details

### A.1  Asset Statistics

We summarize the information of the articulated asset library we used in this project in Table 4. To ease the workload of the LLM in understanding the manipulatable parts of an Articulation, we only reserve one unfixed (prismatic or revolute) joint in each object and fix the others. For rigid body objects, we build a dataset adapted from the YCB object dataset in Maniskill2. We have modified the scale and joint limits of the objects to make the scene proper.

### A.2  Task Generation Statistics

In this section, we investigate the statistics of the tasks in the generation process. We have generated 100 tasks based on the asset library described in Appendix A.1, including 50 primitive tasks and 50 long-horizon tasks. Notice that we consider tasks with the same object class but different instances as the same tasks. Despite the tasks we have generated, these tasks do not include all assets in the asset library and the asset library can also be expanded to contain more object classes and instances.

### A.3  Task Solver Details

**kPAM Planner.** kPAM[57] is a keypoint-based planner proposed for category-level manipulation tasks, and we follow the roadmap proposed in Wang et al. [33] to improve kPAM and apply it in our articulation tasks. It generally defines a robotic task by the homogeneous transformation required to manipulate the target object. kPAM discovers such transformation by solving an optimization problem built on several keypoint-based constraints. After kPAM implicitly defines the end-effector pose to get contact with the object, which we call the *actuation pose*, we need to build the trajectory to approach the actuation pose and subsequently complete the task. To this end, *pre-actuation* and *post-actuation motions* are introduced, both of which include a series of waypoints to guide the end-effector motions (such as making a turn or pushing by moving forward). Together with the actuation pose, an object-centric trajectory of key poses is designed to solve an articulation manipulation task. We provide an illustration of leveraging kPAM planner to solve a task in Fig. 8.

**RL Learner.** In addition to planners, we also leverage a reinforcement learning (RL) learner [10] as an alternative solver for the generated manipulation tasks, in which we prompt an LLM to produce a reward function for a target task. After the task proposal stage, we're provided a task definition and task code generated by the LLM which is used as context for a LLM to generate an executable reward function. The LLM is supplemented with a small library of curated reward examples as well as an API consisting of reward functions corresponding to various predefined components that it can refer to, such as the distance between the end-effector and the object, or the joint positions of certain articulated objects. If successful generation is achieved, we append the reward into the task code and execute the final learning code using Proximal Policy Optimization (PPO)[66] and fix a default set of hyperparameters for all tasks.

In summary, we propose the kPAM motion planner and RL policy learner as complementary methods for solving generated tasks. For kPAM, since it is defined by a well-formulated optimization problem regarding fine-grained constraints and costs, the output motions are much smoother and more natural than motions from other learning-based methods. Moreover, kPAM is inherently generalizable to different scene configurations and object instances because the constraints and actuation motions are all "object-centric" elements, which will be further exploited for domain randomization in Section 3.3. Furthermore, kPAM is fast to execute ($\sim 2$ seconds planning time given a configuration) and robust to multiple runs. Despite the strength of kPAM, RL provides advantages in creative solution proposals for tasks that are too complex or ambiguous for kPAM to define proper constraints, especially for those involving thin objects or contact-rich manipulation. In this project, we give priority to the kPAM planner in most cases because it produces more realistic and generalizable motions, and leaves those that can not be solved for the RL learner.

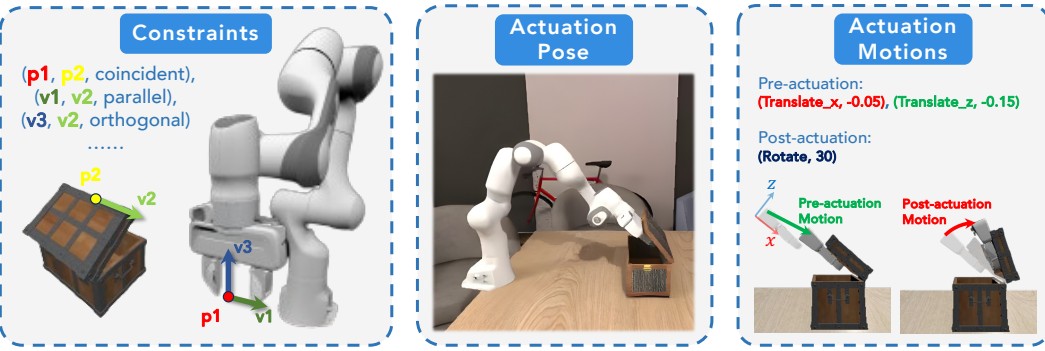

*Figure 8.* **Illustration of kPAM Planner.** We show how to leverage the kPAM planner to solve the task OpenBox. First, constraints are defined to make the gripper get contact with the box lid. Then based on the this actuation pose, some actuation motions are assigned to complete the motion of opening a box.

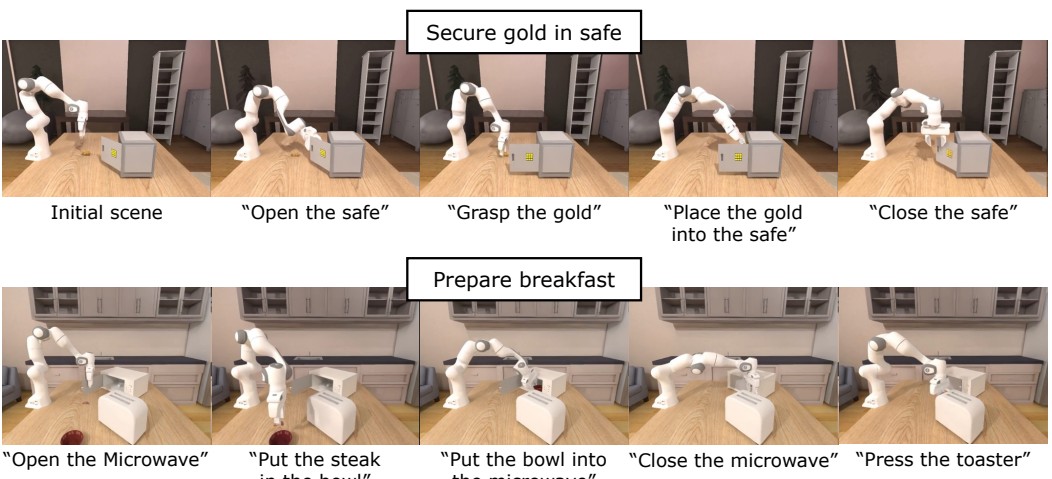

*Figure 9.* **Illustration of Generated Long-horizon Tasks.** We show two example tasks generated by our method, including securing gold in a safe (top) and preparing breakfast (bottom). The former contains 4 sub-tasks and is solved by chaining motion planners of all sub-tasks. The latter contains 7 sub-tasks and is teleoperated due to its complexity.

## A.4  Human Verification

We measure the human efforts required in `GenSim2`, including labeling keypoints for the motion planner and multi-shot rejection sampling in task solver generation.

*Keypoint Labeling.* We set up a protocol for keypoint labeling and annotated 1-3 keypoints for each asset. We measured the time spent on 10 objects and the average time is 8.2s per task.

*Rejection Sampling.* While task solver generation inevitably involves some degree of human intervention, we have made significant efforts to minimize this effort. To evaluate the efficiency of our approach, we conducted a user study measuring the time required for a human user to generate a novel task and corresponding solver using `GenSim2`, and compared it with the time needed for manual task design. The results are presented in Tab. 3. We consider the performance of one of the project's authors (User 1) as expert proficiency and additionally involve two other users who are unfamiliar with our pipeline. The findings indicate that our method significantly reduces task creation time by 50%, and over 90% of the required time involves waiting for LLM to response. Furthermore, the reduced performance gap between experts and beginners suggests that our pipeline requires less technical expertise, making it accessible to users with varying levels of familiarity.

*Table 3.* **Task creation time measurement.** The results are all based on 5 tasks. The average task creation time is around 4 minutes for `GenSim2`.

| Method | User 1 (Expert) | | User 2 | | User 3 | |
|---|---|---|---|---|---|---|
| | GenSim2 | Manual | GenSim2 | Manual | GenSim2 | Manual |
| Average | **159.4s** | 463.8s | **241.0s** | 778.6 | **292.0s** | 727.4s |

## A.5 Demonstration Generation Statistics

In this section, we assess the efficiency and quality of demonstration generation using our kPAM-based motion planner. Recalling our goal to use our generation pipeline as a data amplifier to enhance the generalization and sim-to-real transfer ability of a simulation policy, the generated data should be diverse and of high quality for data augmentation or domain randomization.

To this end, we evaluate the robustness of our kPAM solver to random scene configurations and object instances. For primitive tasks, we add uniform noise in $[-0.1m, 0.1m]$ to the x and y positions of the object, as well as a random rotation in $[-30°, 30°]$ to the yaw axis. For some hard long-horizon tasks, in which the articulation is so small that it may be struggling to place something inside, we fail to generate successful demonstrations with normal randomization, so we reduce the scale of noise by half. For articulations, we randomly select an instance adapted from PartNet-Mobility and add noise to its default joint position.

We test the success rates of generating demonstrations for 24 single-step and 15 long-horizon tasks, and report them task by task in Tab. 5. For each task, we run the demo collection process 5 times, and each time the kPAM solver is executed for 50 episodes with both spatial and object randomization. We compute the mean and standard deviation of the 5 runs on the success rate.

Table 4. Statistics of assets used in `GenSim2`

| Object | Num | Joint Type | Asset Description |
|---|---|---|---|
| laptop_rotate | 44 | revolute | A laptop fixed on the table, with a lid connected by a revolute joint that can be opened and closed |
| stapler_move | 3 | prismatic | A stapler that can be moved on the table |
| window_push | 2 | prismatic | A window with a frame connected by a prismatic joint that can be pushed left and right |
| suitcase_move | 3 | prismatic | A suitcase that can be moved on the table |
| oven | 4 | revolute | An oven fixed on the table, with a door connected by a revolute joint that can be opened and closed |
| drawer | 20 | prismatic | A drawer fixed on the table, connected with its body by a prismatic joint that can be opened and closed |
| box_move | 1 | prismatic | A box that can be moved on the table |
| kitchen_pot | 3 | prismatic | A kitchen bot fixed on the table, connecting its lid and body with a prismatic joint |
| bucket_lift | 5 | prismatic | A bucket that can be lifted from the table |
| trash_can | 3 | revolute | A trash can fixed on the table, with a lid connected by a revolute joint that can be opened and closed |
| stapler_press | 3 | revolute | A stapler fixed on the table, with its handle connected to its body by a revolute joint that can be pressed |
| bottle | 2 | revolute | A bottle fixed on the table, with a top connected to its body by a prismatic joint |
| bag_move | 2 | prismatic | A bag that can be moved on the table |
| dishwasher | 3 | revolute | A dishwasher with a door connected by a revolute joint that can be opened and closed |
| suitcase_rotate | 4 | revolute | A suitcase fixed on the table, with a lid connected by a revolute joint that can be opened and closed |
| toaster_move | 10 | prismatic | A toaster that can be moved on the table |
| bag_swing | 2 | revolute | A bag fixed on the table, with a strap connected by a revolute joint |
| toilet | 2 | revolute | A toilet fixed on the table, lid connected by a revolute joint |
| door | 2 | revolute | A door with its frame connected by a revolute joint that can be opened and closed |
| coffee_machine | 2 | prismatic | A coffee machine fixed on the table, with a button connected by a prismatic joint to be pressed |
| faucet | 13 | revolute | A faucet fixed on the table, with a handle connected by a revolute joint that can be turned on and off |
| washing_machine | 2 | revolute | A washing machine fixed on the table, with a door connected by a revolute joint |
| switch | 2 | revolute | A switch with a frame connected by a revolute joint |
| bucket_swing | 26 | revolute | A bucket fixed on the table, with a handle connected by a revolute joint |
| safe_move | 2 | prismatic | A safe that can be moved on the table |
| toaster_press | 10 | prismatic | A toaster fixed on the table, with a button connected by a prismatic joint |
| window_rotate | 2 | revolute | A window with a frame connected by a revolute joint |
| refrigerator | 2 | revolute | A refrigerator fixed on the table, with a door connected by a revolute joint that can be opened and closed |
| laptop_move | 3 | prismatic | A laptop that can be moved on the table |
| safe_rotate | 15 | revolute | A safe fixed on the table, with a door connected by a revolute joint that can be opened and closed |
| bag_lift | 2 | prismatic | A bag that can be lifted from the table |
| microwave | 10 | revolute | A microwave fixed on the table, with a door connected by a revolute joint that can be opened and closed |
| box_rotate | 13 | revolute | A box fixed on the table, with a lid connected by a revolute joint that can be opened and closed |
| bucket_move | 2 | prismatic | A bucket that can be moved on the table |

*Table 5.* Success Rates of demonstration generation on different tasks

| Task Name | Task Type | Success Rate |
|---|---|---|
| OpenBox | Primitive | $0.84 \pm 0.07$ |
| CloseBox | Primitive | $0.94 \pm 0.03$ |
| OpenLaptop | Primitive | $0.76 \pm 0.03$ |
| CloseLaptop | Primitive | $0.95 \pm 0.01$ |
| TurnOnFaucet | Primitive | $0.67 \pm 0.05$ |
| TurnOffFaucet | Primitive | $0.72 \pm 0.03$ |
| OpenDrawer | Primitive | $0.80 \pm 0.02$ |
| PushDrawerClose | Primitive | $0.87 \pm 0.06$ |
| SwingBucketHandle | Primitive | $0.89 \pm 0.03$ |
| PressToasterLever | Primitive | $0.96 \pm 0.03$ |
| RotateMicrowaveDoor | Primitive | $0.92 \pm 0.01$ |
| CloseSafe | Primitive | $0.80 \pm 0.04$ |
| OpenSafe | Primitive | $0.62 \pm 0.03$ |
| PushToasterForward | Primitive | $0.99 \pm 0.01$ |
| CloseSuitcaseLid | Primitive | $0.82 \pm 0.05$ |
| SwingSuitcaseLidOpen | Primitive | $0.81 \pm 0.06$ |
| RelocateSuitcase | Primitive | $0.80 \pm 0.04$ |
| LiftBucketUpright | Primitive | $0.76 \pm 0.05$ |
| MoveBagForward | Primitive | $0.72 \pm 0.05$ |
| CloseMicrowave | Primitive | $0.53 \pm 0.04$ |
| SwingDoorOpen | Primitive | $0.79 \pm 0.03$ |
| ToggleDoorClose | Primitive | $0.80 \pm 0.05$ |
| CloseRefrigeratorDoor | Primitive | $0.82 \pm 0.03$ |
| OpenRefrigeratorDoor | Primitive | $0.75 \pm 0.06$ |
| PlaceGolfBallIntoDrawer | Long-horizon | $0.56 \pm 0.09$ |
| PlaceCrackerBoxIntoDrawer | Long-horizon | $0.53 \pm 0.08$ |
| PlaceLemonIntoDrawer | Long-horizon | $0.53 \pm 0.06$ |
| PlaceSoftBallIntoDrawer | Long-horizon | $0.58 \pm 0.06$ |
| DropAppleIntoDrawer | Long-horizon | $0.58 \pm 0.05$ |
| StachCupInBox | Long-horizon | $0.40 \pm 0.09$ |
| PlaceGolfBallIntoBox | Long-horizon | $0.58 \pm 0.06$ |
| PutCrackerBoxInBox | Long-horizon | $0.48 \pm 0.08$ |
| StoreBlockInBox | Long-horizon | $0.62 \pm 0.03$ |
| DropAppleIntoBox | Long-horizon | $0.64 \pm 0.04$ |
| StoreLemonInRefrigerator | Long-horizon | $0.48 \pm 0.07$ |
| PlaceCrackerBoxIntoRefrigerator | Long-horizon | $0.26 \pm 0.09$ |
| SecureGoldInSafe | Long-horizon | $0.21 \pm 0.05$ |
| FillMugWithWater | Long-horizon | $0.13 \pm 0.04$ |

## B  Multi-Task Training Details

### B.1  Task Settings

We gradually add the number of training tasks from 4 to 24 in Section 5.1 and test the scaling ability of our framework on the original 4 tasks. We provide a detailed task list here to better clarify our task settings:

- *4 tasks*: OpenBox, CloseBox, OpenLaptop, CloseLaptop;

- *10 tasks*: *4 tasks* + OpenDrawer, PushDrawerClose, SwingBucketHandle, LiftBucketUpright, PressToasterLever, PushToasterForward;

- *15 tasks*: *10 tasks* + MoveBagForward, OpenSafe, CloseSafe, RotateMicrowaveDoor, CloseMicrowave;

- *20 tasks*: *15 tasks* + CloseSuitcaseLid, SwingSuitcaseLidOpen, RelocateSuitcase, TurnOnFaucet, TurnOffFaucet;

- *24 tasks*: *20 tasks* + SwingDoorOpen, ToggleDoorClose, CloseRefrigeratorDoor, OpenRefrigeratorDoor.

### B.2  Architecture Implementation

We implemented a multi-modal policy architecture, as illustrated in Fig. 4, which includes a transformer policy stem, an action head, and different encoders for modeling various types of observations as tokens.

Specifically, we take the CLIP [67] tokenizer to encode the task instruction, which is frozen during training; a pre-trained PointNext [68] to encode the point cloud, which is fine-tuned during training; an MLP for encoding the proprioception states, which is trained from scratch. The transformer conducts self-attention over all tokens, and then we post-process the tokens for different action heads. For example, we compute the mean pooling of all the tokens as the global condition of the diffusion and MLP head; for the transformer, we compute cross-attention between the modeled token and a set of position embeddings to get the final action sequence. Regarding the PointNext encoder, we utilize the pre-trained model on ScanObjectNN Classification[1], as it does not include color information, making it easier for downstream sim-to-real transfer. Note that the diffusion head and the transformer head both model and predict action sequences, but the MLP head only works for single-step action.

### B.3  Policy Architecture Comparison against Baselines.

We first evaluate the Proprioception Pointcloud Transformer (PPT) in solving multiple tasks. We train one policy with demonstrations of 10 tasks on the commonly used manipulation benchmark, RLBench [32]. In particular, we train PPT for 250 epochs on data collected using the same setup as in Yan et al. [69]. Results shown in Tab. 6 performance best performances over 6/10 tasks improved performance over recent competitive baseline methods such as PerAct [46] and GNFactor [70] with a significant reduction in required camera views and pre-trained feature representations. Note that GNFactor [70] requires rather hassles to set up for real-world experiments, as it requires getting features from the neural radiance field [71]. On the contrary, our proposed PPT architecture only requires point cloud, languages, and robot sensor states, which are much easier to obtain with real-world robots.

### B.4  Additional Experiments for Object-Level Generalization

After multi-task training, we test the generalization ability of our policy on unseen object instances. We chose tasks using assets with more than 20 instances and 10 instances to respectively construct

---

[1]https://drive.google.com/drive/folders/1A584C9x5uAqppbjNNiVqlA_
7uOOOlEII?usp=sharing

*Table 6.* **Multi-Task Performance on RLBench.** We evaluate 25 episodes for each checkpoint on 10 tasks across 3 seeds and report the success rates (%) of the final checkpoints. Our method outperforms the most competitive baseline PerAct [46] and GNFactor [70] over 6/10 tasks.

| Methods | turn tap | drag stick | open fridge | put in drawer | sweep to dustpan | meat off grill | phone on base | place wine | slide block | put in safe |
|---|---|---|---|---|---|---|---|---|---|---|
| PerAct | $57.3_{\pm6.1}$ | $14.7_{\pm6.1}$ | $4.0_{\pm6.9}$ | $16.0_{\pm13.9}$ | $4.0_{\pm6.9}$ | $77.3_{\pm14.0}$ | $\mathbf{98.7_{\pm2.3}}$ | $6.7_{\pm6.1}$ | $\mathbf{29.3_{\pm22.0}}$ | $48.0_{\pm26.2}$ |
| GNFactor | $56.0_{\pm14.4}$ | $68.0_{\pm38.6}$ | $2.7_{\pm4.6}$ | $12.0_{\pm6.9}$ | $\mathbf{61.3_{\pm6.1}}$ | $\mathbf{77.3_{\pm9.2}}$ | $96.0_{\pm6.9}$ | $8.0_{\pm6.9}$ | $21.3_{\pm6.1}$ | $30.7_{\pm6.1}$ |
| PPT (ours) | $\mathbf{68.0_{\pm0.10}}$ | $\mathbf{69.3_{\pm0.06}}$ | $\mathbf{10.1_{\pm0.10}}$ | $\mathbf{22.7_{\pm0.06}}$ | $50.1_{\pm0.06}$ | $57.3_{\pm0.02}$ | $44.0_{\pm0.04}$ | $\mathbf{29.3_{\pm0.06}}$ | $21.3_{\pm0.06}$ | $\mathbf{52.0_{\pm0.00}}$ |

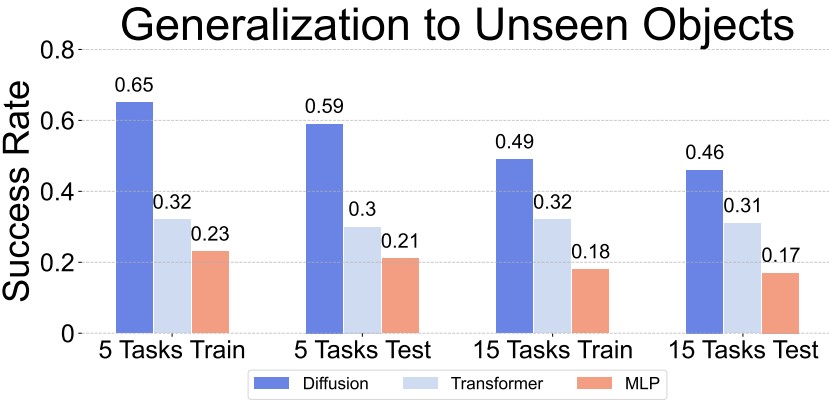

*Figure 10.* Results for object-level generalization test. We show that our policies with various head architectures can generalize to unseen objects.

two training sets of 5 tasks and 15 tasks. During training, we only leverage data generated from 90% of the instances and leave the remaining 10% for testing. The initial poses of the objects are randomly initialized in both procedures as mentioned.

We find that the success rates only drop by around 3% on unseen instances, as shown in Fig. 10. These results indicate that by generating data with object-level and spatial-level variance as domain randomization, our policy can acquire generalization in both aspects, which lays the foundation for further sim-to-real transfer.

## C  Real-World Details

We selected 8 real-world tasks to evaluate the multi-task policy trained on generated data from our pipeline. The *OpenLaptop* task requires the robot to fully open a partially closed laptop, similarly *CloseLaptop* closes the laptop lid. The *OpenSafe* and *CloseSafe* performed similar opening and closing motions on an articulated safe door. The *CloseDrawer* task required the robot to reach and close an open drawer. The generated *SwingBucket* task required the robot to push a handle perpendicular to its current position. The *OpenBox* task required the robot to open a box lid from a partially closed position and the *CloseBox* task required the robot to fully close a box lid.

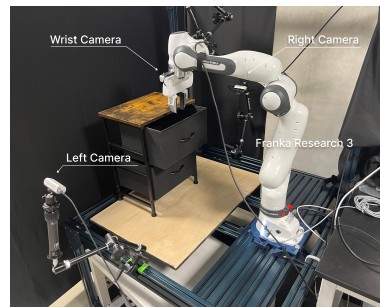

*Figure 11.* Real-world Setup.

As we didn't utilize a discrete action space such as keyframes for real-world experiments, real-time control was important. We achieved fast inference speeds with our multi-task policy with 0.1s inference latency on an NVIDIA 3080 GPU. We parallelized and synchronized point cloud processing from 3 Intel Realsense D435 cameras in order to prevent additional latency. Please view Fig. 11 to see the robot workcell described. Processing the point clouds involved a uniform sampling

*Table 7.* **Real-world Generalizability Experiments.** We transfer a multitask policy from sim to real and evaluate 15 episodes for each task and object and report the success rates below.

| Task | OpenBox | | | | OpenLaptop | | CloseDrawer | | **Average** |
|---|---|---|---|---|---|---|---|---|---|
| Asset | Small | Medium | Long | Large | Toy | Real | Short | Tall | |
| Success Rate | 46.7% | 60.0% | 40.0% | 40.0% | 80.0% | 73.3% | 80.0% | 53.3% | 61.7% |

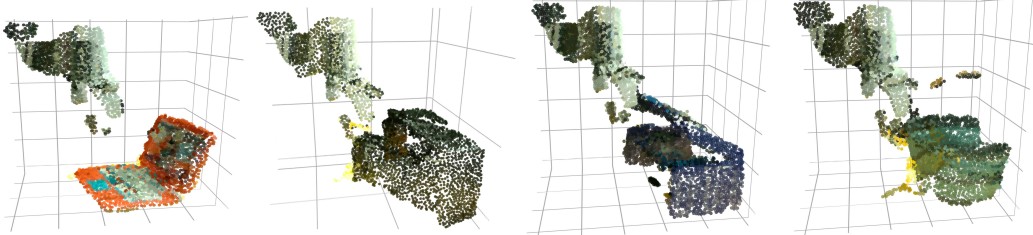

*Figure 12.* **Real World 3D Observation:** Point clouds of different objects in the real world. Category of objects from left to right: laptop, safe, box, and bucket. Color is shown for easier understanding, however, color is not used as input to the multi-task policy.

step for time efficiency, farthest point sampling, and outlier removal to clean and simplify the point cloud for successful inference during Sim-to-Real transfer.

### C.1 Generalization on Unseen Objects

In addition, we also evaluate how well the multi-task policy could generalize to different objects within each task. We run an additional set of experiments for the *OpenBox*, *CloseDrawer*, and *OpenLaptop* tasks, on varying objects. Thanks to the training scale in simulation and the design of our PPT policy architecture, our method can achieve over 60% success rate on multiple tasks with various unseen objects, as shown in Tab. 7. In addition, the performances on different objects within the same task are comparable, demonstrating the robust real-world generalization achieved by our policy.

### C.2 Co-Training Experiment Details

To conduct additional experiments on different multi-task policies incorporating real-world data we incorporated a teleoperation setup using an HTC Vive controller and base station to track 6-DOF hand movements which were mapped to control the robot arm's pose. To interface with the HTC Vive controller we used the (triad-openvr package) and SteamVR. We implemented a position-velocity controller for the robot to track given poses from the controller. Using this teleoperation system, we col-

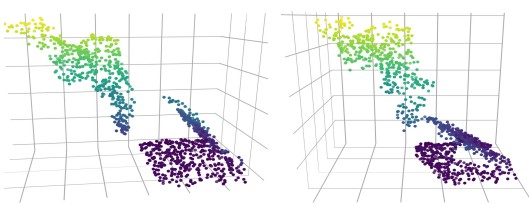

*Figure 13.* Comparison of simulation (left) vs real (right) pointcloud of a laptop.

lected 10 demonstrations for each generated task, collecting state and observation data for each trajectory. We co-trained a multi-task policy using 100 episodes of simulation data per task and the 10 collected demonstrations of real data per task.

## D Comparison with Previous Works

Different from previous efforts including GenSim (top-down manipulation tasks), RoboGen (dynamic tasks), and Robocasa (mobile manipulation tasks with multiple embodiments), GenSim2 focuses on 6-DOF realistic tasks that allow task and high-quality data generation at scale, which really

*Table 8.* Comparison to several released generation frameworks in robotics simulation.

| Methods | **GenSim2 (ours)** | GenSim[9] | RoboGen[10] | RoboCasa[11] |
|---|---|---|---|---|
| Task Type | Articulation | Top-down | Articulation | Articulation |
| Data Generation | From scratch | From scratch | From scratch | Teleoperated |
| Efficiency | Fast | Fast | Slow (RL) | Slow (teleoperation) |
| Data Transferability | High | High | Low (RL) | Medium |
| Sim-to-Real | Zero-shot/co-train | Zero-shot/finetune | None | Co-train |

benefits solving real-world tasks. However, in contrast to GenSim2, GenSim is different from the task level, which primarily deals with simpler top-down pick-place tasks; RoboGen is different from the demonstration generation method, which leverages RL for manipulation tasks and thus produces unrealistic and untransferable data for real-world tasks; RoboCasa is different from the pipeline, which requires large amounts of human teleoperation to collect source data. Moreover, our experiments on taking advantage of generated simulation data for real-world transfer show that we achieve a success rate of 42.5% for zero-shot sim2real transfer and 57.5% for transfer via co-training with both simulation and real data, while RoboCasa, which conducts similar sim2real experiments as ours, only achieves score lower than 25% by co-training. We summarize features of generated data of the aforementioned previous works in Tab. 8.

Furthermore, our task and data generation method features multimodal LLMs as well as a generalizable planner to solve such complex tasks. In our experiment, we verify that the multi-modal task solver generation in this system can greatly improve the task generation success rates by 60%, outperforming previous work by 20%, and the planner used is also friendly and convenient for data generation.

# E    Prompt Examples

## E.1    Prompt Templates

In this section, we demonstrate some examples of the prompt templates we have used to query LLM. The prompt templates are shown as follows:

---

**Prompt: Task Proposal**

You are an expert in creating robotic simulation environments and tasks. You are given some articulated assets for example. Please come up with a creative use of the gripper to manipulate a single Articulation. Note that the simulation engine does not support deformable objects or accurate collision models for contacts. Moreover, the robot can only execute a simple trajectory of one motion.
===========================
Here are all the assets. Please try to come up with tasks using only these assets.
...
===========================
Here are some examples of good tasks. Try to learn from these structures but avoid overlapping with them.
...
===========================
Here are some bad example task instances with reasons. Try to avoid generating such tasks.
...
reasons: ...
===========================
Please describe a NEW task in natural languages and explain its novelty and challenges.
Note:
- Do not use assets that are not in the list above.
- Do not repeat the tasks similar to the good examples or the already generated tasks.
- The task needs to obey physics and remain feasible.
- Do not create similar tasks with the same "assets-used" set.
- All the assets are on the table when initialized.
- Do not place objects on small objects.
- Only one Articulation can be loaded.
- The task contains a simple trajectory of one motion.
- The task should have a clear goal, e.g. use "open/close" instead of "adjust position".

Before the next step, please check if the generated task is a bad task shown in the above examples and meets all the criteria as stated above. Specifically, if the task **only** contains a simple trajectory of one motion, and should have a **clear** goal. Explain in detail, and get a conclusion. If the task is a bad task, regenerate a new one.

Then, format the answer in a Python dictionary with keys "task-name" and value type string, "task-description" (one short phrase), and value type string with lower-case and separated by hyphens, "assets-used" and value type list of strings, and "success-criteria" (choose from "articulated_open", "articulated_closed", "distance_articulated_rigidbody", "distance_gripper_rigidbody", and "distance_gripper_articulated") and value type list of strings. Try to be as creative as possible.

Please remember not to add any extra comments to the Python dictionary.
Let's think step by step.

---

**Prompt: Task Decomposition**

You are an expert in creating robotic simulation environments and tasks. A robot arm with a 2-finger gripper is used in all the robotic simulation environments and tasks. In each task, there is exactly one Articulation and one rigid body object that you can manipulate. You will be given long-horizon tasks with each task including at least 2 sub-tasks. Each sub-task can only include one simple motion such as moving the gripper to some object, opening or closing the gripper fingers, or interacting with certain Articulations by its prismatic/revolute joints.

Please come up with a decomposition of the given long-horizon task to get several sub-tasks. Some rules of such decomposition are listed here:

---

1. Each long-horizon task can not include over 5 sub-tasks, and usually 3-4 are enough.
2. Each sub-task should only include one simple motion as mentioned.
3. Each sub-task(except "grasp" and "ungrasp") should be presented in the format of a Python dictionary with keys "task-name" and value type string with lower-case and separated by hyphens, "task-description" (one specific sentence) and value type string, "assets-used" and value type list with necessary asset(s) in the current sub-task, and "success-criteria" (choose from "articulated_open", "articulated_closed", "distance_articulated_rigidbody", "distance_gripper_rigidbody", and "distance_gripper_articulated") and value type list of strings.
4. Each sub-task should have only one asset used in the task.
5. If the motion of opening or closing the gripper fingers is included in the whole task, it should be listed as a separate sub-task, whose "task-name" should strictly be "grasp" or "ungrasp" respectively and should be the only key in the dictionary.
===========================
Here is an example of the decomposition of the following long-horizon task "...":

...
# Sub-task 1
...
# Sub-task 2
...
# Sub-task 3
...
# Sub-task 4
...
===========================
Now please start to generate a sub-task decomposition of the following new task:

...

## Prompt: Code Generation

Now I will provide you with some reference code and you can write the code for the task "TASK_NAME_TEMPLATE".

...
===========================
The generated code should follow the same structure as the reference and call similar functions.
Do not use libraries, extra functions, properties, arguments, or assets that you don't know.
Remember to import used functions from the corresponding files as the example task codes.
For the Articulation, use "self.articulator" to refer to it, which should be the same as the "assets-used" of the task.
For the objects used, you only have to pass the corresponding parameter (e.g, articulator) and its name (string format, e.g., 'box') in the '__init__' function as shown in above codes, and the base class will automatically load them.
Please comment on the code to explain what each piece does and why it's written that way.

Now write the code for the task "TASK_NAME_TEMPLATE" in the Python code block.

## Prompt: kPAM Solver Stage 1

You are an expert in solving robotic tasks by coding task solution configs. Now please solve the newly generated task by generating the task solution config.

The task solution config contains the necessary positions, parameters, and keypoints for an existing trajectory optimization algorithm to solve a feasible solution. It mainly contains two parts, constraints and pre/post-actuation motions:
(1) The constraints are used to ensure the gripper is in contact with the object and to implicitly define a certain actuation pose.
(2) The pre-actuation motions are used to move the gripper to the actuation pose, while the post-actuation motions are used to complete the task after the actuation pose.

===========================
Here is the task description.
...
===========================

Here are all the available keypoint names for the used manipulator and asset and their descriptions.
...
===========================
Here are some examples of the constraint part of some configs.
...
===========================
Note that, in the constraint list, you need to define different items of constraint to define an actuation pose for the task. There are some pre-defined types of constraints you can use:
(1) point2point_constraint: This constraint is used to ensure two keypoints ("keypoint_name" and "target_keypoint_name", respectively on the tool and object) are in contact.
(2) frame_axis_parallel: This constraint is used to ensure two axes (respectively on the tool and object) are parallel. The axis on the tool is defined by a unit vector from "axis_from_keypoint_name" to "axis_to_keypoint_name", while the axis on the object is defined by "target_axis"([1,0,0] or [0,1,0] or [0,0,1]) which is in the coordinate frame of "target_axis_frame"(world or object).
(3) frame_axis_orthogonal: This constraint is used to ensure two axes (respectively on the tool and object) are orthogonal. The axis on the tool is defined by a unit vector from "axis_from_keypoint_name" to "axis_to_keypoint_name", while the axis on the object is defined by "target_axis"([1,0,0] or [0,1,0] or [0,0,1]) which is in the coordinate frame of "target_axis_frame"(world or object).
(4) keypoint_axis_parallel: This constraint is used to ensure two axes (respectively on the tool and object) are parallel. The axis on the tool is defined by a unit vector from "axis_from_keypoint_name" to "axis_to_keypoint_name", while the axis on the object is defined by another unit vector from "target_axis_from_keypoint_name" to "target_axis_to_keypoint_name".
(5) keypoint_axis_orthogonal: This constraint is used to ensure two axes (respectively on the tool and object) are orthogonal. The axis on the tool is defined by a unit vector from "axis_from_keypoint_name" to "axis_to_keypoint_name", while the axis on the object is defined by another unit vector from "target_axis_from_keypoint_name" to "target_axis_to_keypoint_name".
The tolerance is used to define the tolerance of the constraint.
The target_inner_product is used to define the inner product between the two axes. For example, if you want to ensure two axes are parallel and in the same direction, you can set the target_inner_product to 1.0. If you want to ensure two axes are parallel and of opposite directions, you can set the target_inner_product to -1.0. If you want to ensure two axes to be orthogonal, you can set the target_inner_product to 0.0.

Usually, you need to define one point2point_constraint to ensure contact and several axis constraints to adjust the actuation pose.
===========================
Now please first generate the constraint part for task "TASK_NAME_TEMPLATE" in the same config format as the above.
Do not use terms that you have not seen before.
The output should be in the YAML format with no extra text.

## Prompt: kPAM Solver Stage 2

You are an expert in solving robotic tasks by providing some motion plans and coding task solution configs for a 2-finger robot arm. Now please solve the newly generated task by generating the task solution config.

The task solution config contains the necessary positions, parameters, and keypoints for an existing trajectory optimization algorithm to solve a feasible solution. It mainly contains two parts, constraints and pre/post-actuation motions:
(1) The constraints are used to ensure the gripper is in contact with the object and to implicitly define a certain actuation pose. An actuation pose means the key frame that the robot arm manipulates the object, usually representing the moment when the gripper gets contact with the object.
(2) The pre-actuation motions are used to move the gripper to the actuation pose, while the post-actuation motions are used to complete the task after the actuation pose.
===========================
Here is the task description.
...
===========================
Here are all the available keypoint names for the used manipulator and asset and their descriptions.
...
===========================

Here are some examples of the pre/post-actuation part of some task solution configs.

...

===========================

Now please generate SOLVER_TRIALS different pre/post-actuation motions for task "TASK_NAME_TEMPLATE" following the same config format shown above based on the constraint part that is generated previously. The pre/post-actuation motions of different solutions can be diverse, but their task names should be the same.

Do not use terms that you have not seen before.

The output should be in the YAML format with no extra text.

The diversity of the motions can be achieved by using different axes for translation.

Notice that the pre-actuation and post-actuation motions are relative to the actuation pose and the translation motions are represented in coordinates relative to the manipulator base.

Let's think step by step, and try your best to understand the job.

## E.2 Example for Multi-modal LLM Response

We provide an example conversation when generating a solver config for the task "close-drawer" with a multi-modal LLM, and compare it with the response generated by a vanilla LLM. Some of the prompt details are omitted, and please refer to Appendix E for the complete prompts.

---

**Example for Multi-modal LLM Response**

*Input:*

You are an expert in solving robotic tasks by coding task solution configs. Now please solve the newly generated task by generating the task solution config.

......

```
===========================
Here is the task description.
{
    "task-name": "close-drawer",
    "task-description": "push a drawer to a closed position",
    "assets-used": ["drawer"],
    "success-criteria": ["articulated_closed"]
}
===========================
```

......

The initial scene of this task is visualized in the first uploaded image. The frame axes of the gripper and the object are respectively visulized in the second and third images, where red, green and blue axes represent X, Y and Z axes respectively.

......

Now please first generate the constraint part for task "close-drawer" in the same config format as the above.

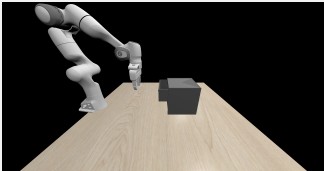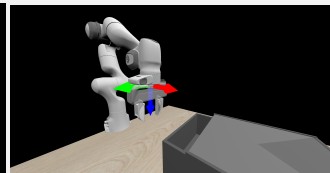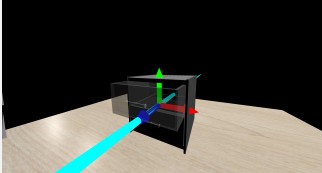

*Output:*

......

task_name: CloseDrawer
category_name: Articulated

tool_keypoint_name_list: [tool_head, tool_tail, tool_side]
object_keypoint_name_list: [articulated_object_head]

constraint_list:
- keypoint_name: tool_head
  target_keypoint_name: articulated_object_head
  tolerance: 0.0001
  type: point2point_constraint
- axis_from_keypoint_name: tool_head
  axis_to_keypoint_name: tool_side
  target_axis: [1.0, 0, 0]

---

```
   target_axis_frame: object
   tolerance: 0.01
   target_inner_product: 1
   type: frame_axis_parallel
- axis_from_keypoint_name: tool_head
   axis_to_keypoint_name: tool_tail
   target_axis: [0, 0, 1.0]
   target_axis_frame: object
   tolerance: 0.01
   target_inner_product: 0
   type: frame_axis_orthogonal
```

*Input:*

......

The initial scene of this task is visualized in the first uploaded image. The actuation pose of this task is visualized in the second uploaded image. The frame axes of the gripper and the object are respectively visualized in the third and fourth images, where red, green, and blue axes represent the X, Y, and Z axes respectively.

......

Now please generate 3 different pre/post-actuation motions for task "close-drawer" following the same config format shown above based on the constraint part that was generated previously. The pre/post-actuation motions of different solutions can be diverse, but their task names should be the same.

......

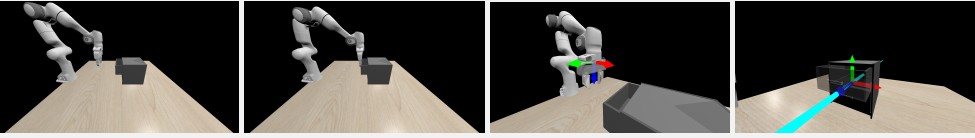

*Output:*

......

task_name: CloseDrawer
category_name: Articulated

# Pre-actuation motions for the manipulator.
pre_actuation_motions:
  - ["translate_x", -0.1]
  - ["translate_z", -0.15]

# Post-actuation motions for the manipulator.
post_actuation_motions:
  - ["translate_x", 0.1]

*Success!*

***Input:***

You are an expert in solving robotic tasks by coding task solution configs. Now please solve the newly generated task by generating the task solution config.

......

```
==========================
Here is the task description.
{
    "task-name": "close-drawer",
    "task-description": "push a drawer to a closed position",
    "assets-used": ["drawer"],
    "success-criteria": ["articulated_closed"]
}
==========================
```

......

Now please first generate the constraint part for task "close-drawer" in the same config format as the above.

***Output:***

......

task_name: CloseDrawer
category_name: Articulated

tool_keypoint_name_list: [tool_head, tool_tail, tool_side]
object_keypoint_name_list: [articulated_object_head]

constraint_list:
- keypoint_name: tool_head
  target_keypoint_name: articulated_object_head
  tolerance: 0.0001
  type: point2point_constraint


- axis_from_keypoint_name: tool_side
  axis_to_keypoint_name: tool_tail
  target_axis: [0, 1.0, 0]
  target_axis_frame: world
  tolerance: 0.01
  target_inner_product: 1
  type: frame_axis_parallel
- axis_from_keypoint_name: tool_head
  axis_to_keypoint_name: tool_tail
  target_axis: [0, 0, 1.0]
  target_axis_frame: world
  tolerance: 0.01
  target_inner_product: 0
  type: frame_axis_orthogonal


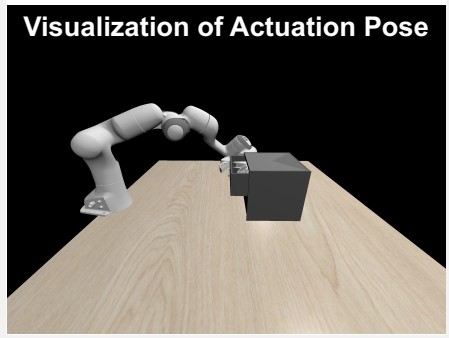

**Visualization of Actuation Pose**

***Fail to generate correct constraints!***

## E.3   Example for Reasoning LLM Response

We provide an example conversation when generating a solver config for the task "close-safe" with a reasoning LLM, and compare it with the response generated by a vanilla LLM. Some of the prompt details are omitted, and please refer to Appendix E for the complete prompts. We highlight some limitations of applying vision language foundation models to robotic simulation today, but we expect further progress, such as finetuning APIs, would be able to correct some of these failure modes.

---

### Example for Reasoning LLM Response

***Input:***

You are an expert in solving robotic tasks by coding task solution configs. Now please solve the newly generated task by generating the task solution config.

......

===========================
Here is the task description.
{
    "task-name": "close-safe",
    "task-description": "Close the safe door",
    "assets-used": ["safe_rotate"],
    "success-criteria": ["articulated_closed"]
}

===========================

......

Now please first generate the constraint part for task "close-safe" in the same config format as the above.

***Output:*** *(Reasoning parts tracked with blue)*
**Step 1: Understand the Task and Keypoints**
- Task: Close the safe door.
- Manipulator Keypoints:
    - tool_head: Middle point between the two finger tips.
    - tool_tail: Middle point between the two finger bases.
    - tool_side: Tip of the right finger.
- Articulated Object Keypoints:
    - articulated_object_head: Middle point of the safe handle.
    - articulated_object_inside_base: Point inside the safe to drop objects.

**Step 2: Plan Feasible Constraints**
We need to define constraints that can be satisfied and are different from the previous attempts.
Constraints typically include:
1. Point-to-Point Constraint: Ensure contact between a keypoint on the tool and a keypoint on the object.
2. Axis Constraints: Orient the tool relative to the object to facilitate the action.

*Config 1*

*Constraint 1: Point-to-Point Constraint*
- Purpose: Ensure the right finger tip contacts the safe handle.
- Implementation:
- keypoint_name: tool_side
    target_keypoint_name: articulated_object_head
    tolerance: 0.0001
    type: point2point_constraint

*Constraint 2: Frame Axis Parallel Constraint*
- Purpose: Align the gripper's Z-axis with the door's closing direction.

---

- Assumptions:
    - The door closes along the object's X-axis ([1, 0, 0]).
    - The gripper's Z-axis is from tool_tail to tool_head.
- Implementation:
- axis_from_keypoint_name: tool_tail
    axis_to_keypoint_name: tool_head
    target_axis: [1.0, 0, 0]
    target_axis_frame: object
    tolerance: 0.01
    target_inner_product: 1.0
    type: frame_axis_parallel

*Constraint 3: Frame Axis Parallel Constraint*
- Purpose: Align the gripper's X-axis with the door's hinge axis.
- Assumptions:
    - The door's hinge axis is along the object's Z-axis ([0, 0, 1]).
    - The gripper's X-axis is from tool_head to tool_side.
- Implementation:
- axis_from_keypoint_name: tool_head
    axis_to_keypoint_name: tool_side
    target_axis: [0, 0, 1.0]
    target_axis_frame: object
    tolerance: 0.01
    target_inner_product: 1.0
    type: frame_axis_parallel

*Config 1*
task_name: CloseSafe
category_name: Articulated

tool_keypoint_name_list: [tool_head, tool_tail, tool_side]
object_keypoint_name_list: [articulated_object_head, articulated_object_inside_base]

constraint_list:
# Ensure the right finger tip contacts the safe handle
- keypoint_name: tool_side
    target_keypoint_name: articulated_object_head
    tolerance: 0.0001
    type: point2point_constraint

# Align gripper's Z-axis with the door's closing direction
- axis_from_keypoint_name: tool_tail
    axis_to_keypoint_name: tool_head
    target_axis: [1.0, 0, 0]
    target_axis_frame: object
    tolerance: 0.01
    target_inner_product: 1.0
    type: frame_axis_parallel

# Align gripper's X-axis with the door's hinge axis
- axis_from_keypoint_name: tool_head
    axis_to_keypoint_name: tool_side
    target_axis: [0, 0, 1.0]
    target_axis_frame: object
    tolerance: 0.01
    target_inner_product: 1.0
    type: frame_axis_parallel

**Summary**
Right finger tip contacts the safe handle, with the gripper's Z-axis aligned with the door's closing direction, and X-axis aligned with the door's hinge axis.

***Input:***

......

Now please generate 3 different pre-actuation motions for task "close-safe" following the same config format shown above based on the constraint part that is generated previously. The pre-actuation motions of different solutions can be diverse, but their task names should be the same.

......

***Output:***

......

task_name: CloseSafe
category_name: Articulated

# Pre-actuation motions for the manipulator.
# Each motion is represented in the [mode, value] format.
# Mode: translate_x, translate_y, translate_z, or rotate.
# Value: distance in meters or angle in radians.
pre_actuation_motions:
   - ["translate_z", -0.1]     # [lift gripper above handle to approach from above]

***Input:***

......

Now please generate 3 different post-actuation motions for task "close-safe" following the same config format shown above based on the constraint part that is generated previously. The post-actuation motions of different solutions can be diverse, but their task names should be the same.

......

***Output:*** *(Reasoning parts tracked with blue)*

**Step 1: Understand the Task**
We need to generate three different post-actuation motion plans for the task"close-safe". The post-actuation motions are the actions the manipulator performs after reaching the actuation pose to complete the task of closing the safe door.

**Step 2: Consider Possible Motions**
Since the goal is to close the safe door, we need to decide how the manipulator can move to achieve this, using only the allowed motion set:
- Option 1: Push the door forward to close it.

**Step 3: Define the Motions According to the Robot's Coordinate System**
- Forward Direction: Along the red axis; useful for pushing actions.
- Backward Direction: Opposite to the red axis; useful for pulling actions.
- Left Direction: Along the green axis; can be used to swing the door closed if it hinges that way.
- Right Direction: Opposite to the green axis.
- Up/Down Directions: Along the blue axis; may not be as relevant for closing a door but can be included for completeness.

**Step 4: Generate Motion Plans**
*Config 1:* Push the door closed by moving forward.
- Post-actuation motions: ["move-forward"]

**Step 5: Ensure Compliance with Constraints**
- Each motion plan uses only the allowed motions.
- Motions are within the 3-step limit.
 - Directions are defined according to the robot's coordinate system.
- Plans avoid collision with the table (e.g., no downward movements that might cause a collision).

**Step 6: Output the Configs in YAML Format**

task_name: CloseSafe
category_name: Articulated

# Post-actuation motions for the manipulator.
# Post-actuation motions should be in the format of a python list. Each element in the list should be a string with lower-case and separated by hyphens.
# post-actuation pose list
post_actuation_motions: ["move-forward"]        # [push door closed]

*Success!*

## Example for Vanilla LLM Response

***Input:***

You are an expert in solving robotic tasks by coding task solution configs. Now please solve the newly generated task by generating the task solution config.

......

```
==========================
Here is the task description.
{
    "task-name": "close-safe",
    "task-description": "Close the safe door",
    "assets-used": ["safe_rotate"],
    "success-criteria": ["articulated_closed"]
}
==========================
```

......

Now please first generate the constraint part for task "close-drawer" in the same config format as the above.

***Output:***

......

task_name: CloseSafe
category_name: Articulated

tool_keypoint_name_list: [tool_head, tool_tail, tool_side]
object_keypoint_name_list: [articulated_object_head]

constraint_list:
# ensure the gripper's tool_head is in contact with the safe handle - keypoint_name: tool_head
    target_keypoint_name: articulated_object_head
    tolerance: 0.0001
    type: point2point_constraint

# ensure the gripper's top side is aligned with the safe handle's direction (assuming the handle moves in the world's Y direction)
- axis_from_keypoint_name: tool_head
    axis_to_keypoint_name: tool_tail
    target_axis: [0, 1.0, 0]
    target_axis_frame: world
    tolerance: 0.01
    target_inner_product: 1
    type: frame_axis_parallel
# ensure the gripper's front side is perpendicular to the world's Y direction
- axis_from_keypoint_name: tool_tail
    axis_to_keypoint_name: tool_side
    target_axis: [0, 1.0, 0]
    target_axis_frame: world
    tolerance: 0.01
    target_inner_product: 0
    type: frame_axis_orthogonal

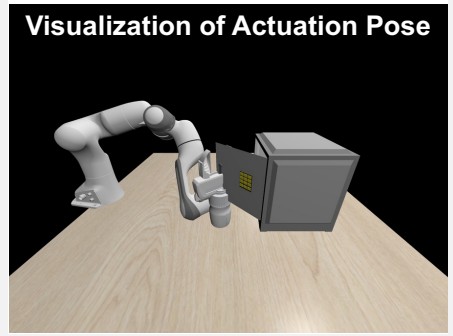

**Visualization of Actuation Pose**

***Fail to generate correct constraints!***

## E.4 Example Failure Cases

We provide some failure cases of different stages in this section.

---

**Failure Case for Task Proposal**

*Input:*

You are an expert in creating robotic simulation environments and tasks. You are given some articulated assets for example. Please come up with creative use of the gripper to manipulate a single articulated object. Note that the simulation engine does not support deformable objects or accurate collision models for contacts. Moreover, the robot can only execute a simple trajectory of one motion.

......

Format the answer in a Python dictionary with keys "task-name" and value type string, "task-description" (one short phrase), and value type string with lower-case and separated by hyphens, "assets-used" and value type list of strings, and "success-criteria" (choose from "articulated_open", "articulated_closed", "distance_articulated_rigidbody", "distance_gripper_rigidbody", and "distance_gripper_articulated") and value type list of strings. Try to be as creative as possible.

......

*Output:*

......

```
{
    "task-name": "unlock-safe",
    "task-description": "rotate the safe's handle to a horizontal position to simulate unlocking",
    "assets-used": ["safe_rotate"],
    "success-criteria": ["articulated_open"]
}
```

*Fail to generate a correct task!*

**Failure reason:** The handle is not rotatable.

---

**Failure Case for Task Decomposition**

*Input:*

You are an expert in creating robotic simulation environments and tasks. You will be given long-horizon tasks with each task including at least 2 sub-tasks. Each sub-task can only include one simple motion such as moving the gripper to some object, opening or closing the gripper fingers, or interacting with certain articulated objects by its prismatic/revolute joints.

......

Now please start to generate a sub-task decomposition of the following new task:

```
{
    "task-name": "tidy-up-foam-brick",
    "task-description": "Pick up the foam brick, open the microwave door, place the brick inside, and close the door.",
    "assets-used": ["microwave", "foam_brick"]
}
```

......

*Output:*

# Sub-task 1
{
 "task-name": "reach-foam-brick",
 "task-description": "reach the foam brick and make the foam brick between gripper fingers",
 "assets-used": ["foam_brick"],
 "success-criteria": ["distance_gripper_rigidbody"]
}

# Sub-task 2
{
 "task-name": "grasp"
}

# Sub-task 3
{
 "task-name": "open-microwave-door",
 "task-description": "open the microwave door by interacting with its revolute joint",
 "assets-used": ["microwave"],
 "success-criteria": ["articulated_open"]
}

# Sub-task 4
{
 "task-name": "place-inside-microwave",
 "task-description": "move the gripper with the foam brick inside the microwave",
 "assets-used": ["microwave"],
 "success-criteria": ["distance_articulated_rigidbody"]
}

# Sub-task 5
{
 "task-name": "ungrasp"
}

# Sub-task 6
{
 "task-name": "close-microwave-door",
 "task-description": "close the microwave door by interacting with its revolute joint",
 "assets-used": ["microwave"],
 "success-criteria": ["articulated_closed"]
}

***Fail to generate a correct task decomposition!***

**Failure reason:** Door should be open before the object is grasped.

---

## Failure Case for Solver Creation

*Input:*

You are an expert in solving robotic tasks by coding task solution configs. Now please solve the newly generated task by generating the task solution config.

......

===========================
Here is the task description.
{
 "task-name": "open-microwave",
 "task-description": "open the microwave door",

"assets-used": ["microwave"],
"success-criteria": ["articulated_open"]
}
============================

......

The initial scene of this task is visualized in the first uploaded image. The frame axes of the gripper and the object are respectively visulized in the second and third images, where red, green and blue axes represent X, Y and Z axes respectively.

......

Now please first generate the constraint part for task "close-drawer" in the same config format as the above.

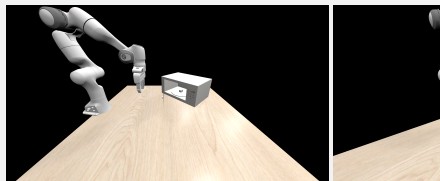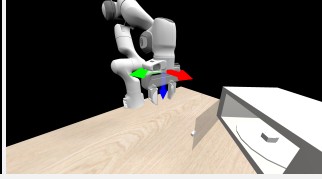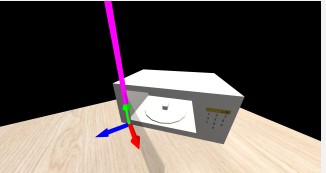

*Output:*

......

task_name: OpenMicrowave
category_name: Articulated

tool_keypoint_name_list: [tool_head, tool_tail, tool_side]
object_keypoint_name_list: [articulated_object_head]

constraint_list:
- keypoint_name: tool_head
  target_keypoint_name: articulated_object_head
  tolerance: 0.0001
  type: point2point_constraint

- axis_from_keypoint_name: tool_head
  axis_to_keypoint_name: tool_tail
  target_axis: [0, 0, 1.0]
  target_axis_frame: world
  tolerance: 0.01
  target_inner_product: 0
  type: frame_axis_orthogonal
- axis_from_keypoint_name: tool_head
  axis_to_keypoint_name: tool_side
  target_axis: [0, 1.0, 0]
  target_axis_frame: object
  tolerance: 0.01
  target_inner_product: 1
  type: frame_axis_parallel

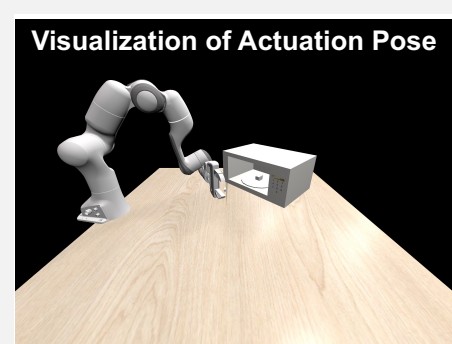

**Visualization of Actuation Pose**

*Fail to generate a correct solver config!*

**Failure reason:** The corresponding actuation pose cannot complete the task.

