# OpenReview forum: "GenSim2: Scaling Robot Data Generation with Multi-modal and Reasoning LLMs"
_robot-learning.org/CoRL/2024/Conference — CoRL 2024_

### Official Review · Reviewer_9Mkg · 2024-07-20
**Utilizing LLMs for Robot Task Generation and Learning**

**Originality:** 3
**Technical Quality:** 3
**Clarity Of Presentation:** 4
**Potential Impact:** 2
**Recommendation:** 2
**Confidence:** 5

**Review:**

The paper is clear and easy to follow, effectively motivating the challenge of automatic task generation and extending it towards long-horizon tasks.

However, a number of components of the work require clarification or additional experimentation. For example:

- **Importance of Visual Reflection**: Section 3.1 mentions that both language and visual reflections of the LLM are used to verify the task, but section 4.1 talks about humans doing the reflection. It might be better to clarify in the methods section that the reflection/feedback can either be provided by humans or LLMs.

   - **Confusion about _GenSim2 w/oV_ results**: The configuration labeled "GenSim2 w/oV" is unclear. The paper describes it as GenSim2 without the vision API of GPT-4, but it is not specified whether this is an ablation of GenSim2-HF or GenSim2-VF. If it is an ablation of GenSim2-VF, it is unclear why removing vision (as in GenSim2 w/oV) would improve performance compared to GenSim2-VF.

- **No compelling examples of Long-Horizon Tasks**: Although generating long-horizon tasks is a key feature of this work, none of the supplementary materials include examples of such tasks. Providing examples would help illustrate the capabilities of the proposed method.

- **Pre-actuation and Post-action Motion**: The paper states that these motions are defined as a series of waypoints. Does each waypoint represent a 6D end-effector pose? How many of such waypoints are specified for each task? Additional details, such as the typical number of waypoints required, should be included in the appendix to provide further clarity.

**Quality Of The Limitations Section:**

3

**Questions For Rebuttal:**

See review comments.

**Robotics Focus:**

4

**Summary Of Paper:**

This work (GenSim2) presents an extension of GenSim, a method for scaling robot task creation using large language models (LLMs). The extensions address increased task complexity, improved task solvers for demonstration data collection, and training a multi-task transformer model. First, the iterative top-down and bottom-up approaches enable the generation of long-horizon tasks involving articulated objects. Second, the integration of a kPAM planner with reinforcement learning (RL) policy learning enhances the efficiency and effectiveness of task-solving for demonstration data collection. Third, the proposed multi-task transformer model is trained to distill the demonstration data into a single policy, demonstrating sim-to-real transfer through the careful selection of observation inputs. The paper shows promising results in generating long-horizon tasks more effectively than previous methods, creating 50 single-step tasks and 50 multi-step tasks. Additionally, the proposed approach solves these tasks and distills multiple tasks into a single policy. Notably, the distilled multi-task policy exhibits effective sim-to-real transfer, primarily due to the use of point cloud observations, which has a small sim-to-real gap.

**Summary Of Recommendation:**

While the work is interesting and addresses the challenge of task generation, it is fairly incremental, and it is not entirely clear if the improvements over GenSim are significant enough to warrant a completely new paper. The tasks remain relatively simple and similar. Additionally, it is unclear how much human input is required for task generation and solving (e.g., human-specified kPAM key points), especially given that the primary motivation for using LLMs in task generation is to reduce the burden on humans in the process.

---

### Official Review · Reviewer_i12f · 2024-07-20
**A solid and functional work with some presentation flaws**

**Originality:** 3
**Technical Quality:** 4
**Clarity Of Presentation:** 3
**Potential Impact:** 3
**Recommendation:** 4
**Confidence:** 3

**Review:**

## Strengths
* **Problem**: given the increasing ability of LLM and VLM in designing tasks, reward functions and policies in simulation, I believe works like this will be important in order to leverage such abilities for sim2real transfer
* **Method**: the framework presents a robust pipeline that should enable effective sim2real transfer with minimal human intervention. The overall methodology and the results are convincing
## Weaknesses
* **Complexity**: the system itself is complex and made of several components, and variables. It is hard to understand which components are contributing to the overall success of the system. The authors presented some ablations, but these don't exhaustively cover all the variables involved (e.g. the simulator, the LLM, the RL policy learner, the motion planner, etc). It would be great if additional ablations could be provided
* **Presentation flaws**: there are some presentation flaws, such as missing information, typos and missing references. I would recommend the authors revise their paper (also considering the suggestions provided in the Questions for rebuttal section)

**Quality Of The Limitations Section:**

2

**Questions For Rebuttal:**

* What is the simulator used? This should be clearly stated in the text (and Figure I caption as well)
* What is the task library adopted? How is this composed/learned?
* List of presentation flaws:
Line 30, GenSim2 1, GenSim2 (Figure 1)
Line 49, 25 -> 25%
Line 110, "language and visual reflections" -> I think a citation to [1] or similar work would be necessary
Paragraph from line 112 to 125, I think the bottom-up approach is very similar to [2] and a citation to it or to similar work would be necessary
Line 196, pcd not defined (pointcloud?)
Line 199, 4.4 -> Section 4.4
Line 230, LLM -> the LLM
Line 231, missing Section (??)

[1] Reflexion: Language Agents with Verbal Reinforcement Learning, Shinn et al

[2] Voyager: An Open-Ended Embodied Agent with Large Language Models, Wang et al

**Post rebuttal update**:

Apologies for the tardiness in the response. I have gone through the extensive rebuttal provided and updated my score to a "Strong Accept".

**Additional ablation studies**: I appreciate the authors' efforts in adding this, as it helps to understand what are the sources of performance improvement in their system.

**Task library**: I think a more detailed description of the task library should be included in the paper, as clarified by the authors in their response.

Overall, given the additional clarifications and the extensive amount of additional content presented by the authors, I think the paper would be a nice addition to the conference, and thus I think it should be accepted.

**Robotics Focus:**

4

**Summary Of Paper:**

The work presents GenSim2 a framework to enable sim2real transfer. GenSim2 leverages LLM and VLMs for designing tasks in simulation, RL and motion planners to obtain demonstrations, and multi-task training, to aid generalization and transfer to real-world applications.

**Summary Of Recommendation:**

A convincing method for sim2real transfer, with minor presentation flaws which are expected to be solvable by the authors.

---

### Official Review · Reviewer_Wb5Y · 2024-07-24
**Paper review**

**Originality:** 3
**Technical Quality:** 3
**Clarity Of Presentation:** 4
**Potential Impact:** 3
**Recommendation:** 3
**Confidence:** 4

**Review:**

Strengths:
1. This paper presents a scalable solution to the important topic of simulation task generation; GenSim2 is compelling with its capability to generate 100+ simulation tasks that go beyond the top-down pick-and-place tasks that prior works are able to generate.

2. This paper designs an effective sim-to-real pipeline with a 3D transformer architecture; this is well executed to show that the generated tasks are usable.

3. The paper is well-written and easy to follow.

Weaknesses:
1. This paper is quite similar to a prior works RoboGen, and on the surface level, it's not clear the novelty of this work in comparison to RoboGen. While there are implementation differences, such as using kPAM solver for demonstration generation (which is rather specific to the task categories this paper considers), the overall system pipeline is rather quite similar.

2. The approach is limited by what is simulatable or available in simulation. That is, this paper takes pre-existing assets and then design the algorithm around them, e.g., the use of kPAM planner which is limited to  object-centric displacement tasks. In contrast, concurrent works like RoboGen or RoboCasa do consider generative new assets.

3. The best performing GenSim2 variant requires human feedback on the generated tasks; this limits the scalability of the approach. Without human feedback, GenSim2 variants do not outperform RoboGen on single-step tasks. On long-horizon tasks, only the variants with human feedback are considered. I think this limitation is rather important, but it's not mentioned in the introduction or the limitation section.

Overall, I think GenSim2 is a sound system but with some important limitations and its comparison against prior works can be strengthened.

**Quality Of The Limitations Section:**

3

**Questions For Rebuttal:**

Additional Question/Comments:
1. In the real-world tasks, some generalization over unseen objects, scenes would be nice to demonstrate that the policies learned in simulation can have good generalization.

2. Typo in line 230.

**Robotics Focus:**

4

**Summary Of Paper:**

This paper proposes GenSim2 a LLM based task generation pipeline that can generate realistic articulated object tasks capable of sim-to-real transfer.

**Summary Of Recommendation:**

Weak Accept because this paper presents a sound system for end-to-end LLM task generation to sim-to-real transfer; however, the technical novelty is rather limited and many components take a lot of engineering to get to work. Furthermore, its reliance on human feedback is not highlighted enough in the paper.

---

### Author Rebuttal · Authors · 2024-08-12

Please refer to our revised paper and updated video in the uploaded zip file.

---

### Decision · Program_Chairs · 2024-09-04

**Decision:**

Accept

**Comment:**

The authors propose an LLM-based task data generation system for robots, together with a 3d-based sim-to-real pipeline. They particularly focused on realistic, 6-dof robot tasks. The work itself seems like a notable improvement over several similar works in this area. Their approach does leverage human expertise when creating tasks, which has advantages and disadvantages, but is interesting compared to the other related works. They also show some ability to generate long-horizon tasks.

During the rebuttal period, they provided a very detailed followup, with information clearly comparing them to related work and showing the ability of models trained using their method to generalize, as well as adding substantial extra discussion that addressed many of the core concerns raised by the reviewers.

Strengths:
	• Paper is mostly well-written and convincing
	• Generates lots of tasks beyond just pick and place - great step towards solving data problems in robotics
	• Good sim-to-real learning pipeline
Weaknesses:
	• Need for human intervention
Generalization is greatly improved from rebuttal, but some concerns remain